# Uptake and Toxicity of Polystyrene NPs in Three Human Cell Lines

**DOI:** 10.3390/ijms26104783

**Published:** 2025-05-16

**Authors:** Sylwia Męczyńska-Wielgosz, Katarzyna Sikorska, Malwina Czerwińska, Lucyna Kapka-Skrzypczak, Marcin Kruszewski

**Affiliations:** 1Centre for Radiobiology and Biological Dosimetry, Institute of Nuclear Chemistry and Technology, 03-195 Warsaw, Poland; s.meczynska@icht.waw.pl (S.M.-W.); k.sikorska@icht.waw.pl (K.S.); m.czerwinska@ichtj.waw.pl (M.C.); 2Department of Molecular Biology and Translational Research, Institute of Rural Health, 20-090 Lublin, Poland; kapka.lucyna@imw.lublin.pl

**Keywords:** nanoplastic, nanoparticle internalization, endocytosis, scavenger receptors

## Abstract

Internalization of nanoparticles (NPs), including nanoplastic, is one of the key factors determining their toxicity. In this work, we studied the toxicity and mechanisms of the uptake of model fluorescent polystyrene NPs (PS NPs) of three different sizes (30, 50, and 100 nm) in three human cancer cells lines; two originated from gut tissue (HT-29 and Caco-2) and one originated from liver tissue (Hep G2). Toxicity was measured by Neutral Red Assay (NRU), whereas mechanisms of uptake were studied using flow cytometry and different uptake inhibitors. The toxicity of the studied NPs followed a general rule observed for NPs—the smaller ones were more toxic than the larger ones. This relationship was dose dependent; however, the overall toxicity of the studied NPs was very low, despite the significant uptake of PS NPs. Although clathrin- and caveolin-dependent uptake is generally accepted as a major route of NP uptake, the inhibition of both mechanisms did not affect PS NP uptake in the cell lines studied in this work. Further experiments revealed that the major route of PS NP uptake in these cells is a scavenger receptor-mediated uptake.

## 1. Introduction

Plastic products that pollute the environment slowly disintegrate into small fragments, eventually forming micro- and nanoplastic particles (MNPlastics), i.e., particles less than 5 mm and 1 µm in size, respectively. These tiny plastic particles of various shapes and sizes are widespread in the environment and can also be found in the human and animal body. Once they have entered the human body, usually by inhalation or digestion [1], the MNPlastics will eventually find their way into cells.

Although internalization mechanisms that are active in a particular cell type likely determine its ability to uptake nanoparticles (NPs), the physicochemical properties of NPs may also affect the process. Several NP features have been proven to affect the internalization process; among them, size, shape, and surface properties are of the greatest importance [2,3]. Furthermore, composition of the protein corona, i.e., external proteins that are bound to the surface of NPs [4], may also affect NP uptake.

The common mechanisms of MNPlastics cellular internalization do not differ substantially from the mechanisms of internalization of other types of NPs. Native NPs are uptaken if they fit natural processes occurring in the cell, such as the uptake of viruses, bacteria. or other antigens, or of natural compounds, e.g., hormones, growth factors, and lipoproteins. Engineered NPs are made to embody natural compounds and merge into those processes, e.g., by linking with ligands that have a high affinity with highly expressed receptors or proteins secreted in tumor microenvironments or by changing other NP surface properties, such as charge, hydrophobicity, etc. [5].

In general, energy-independent and energy-dependent pathways might be considered. The majority of NPs enter cells through energy-dependent mechanism, such as phagocytosis and pinocytosis. Whereas phagocytosis is realized mostly by specialized cells, such as neutrophils or macrophages, pinocytosis is realized by a majority of other cells [6]. Depending on the proteins involved in internalization, pinocytosis is usually further divided into micropinocytosis, clathrin-dependent endocytosis, caveolin-dependent endocytosis, and clathrin- and caveolin-independent endocytosis. Some mechanisms limit the size of uptaken material, e.g., clathrin-coated vesicles are around 100–150 nm in diameter, whereas caveolin-coated vesicles are usually 60–90 nm in diameter. The uptake of NPs by phagocytosis seems to be size independent, at least for NPs with a size of up to 1 μm [7]. Thus, NP properties define the mechanism by which NPs are uptaken.

NP uptake also depends on the target cells. Some specialized cells rely on a particular mechanisms of uptake, e.g., phagocytes and macrophages uptake extracellular material mostly by phagocytosis and micropinocytosis, whereas these mechanisms are rarely utilized by other types of cells (for review see [8,9]). Recently, attention has been paid to the role of receptor-mediated endocytosis in NP internalization. Nowadays, it is clear that NPs can interact with scavenger receptors and be uptaken via receptor-dependent pathways [10,11,12]. Thus, it seems that NP uptake should be considered individually in regard to the particular NP types and cellular systems being investigated.

NP uptake and toxicity depend on similar factors, suggesting a mutual relationship between the uptake and cytotoxicity of NPs. Once NPs reach the plasma membrane, they are uptaken, usually by endocytosis, directed via an endosome trafficking network, and then sorted to different cellular components. Otherwise, NPs or their contents are released into the cytoplasm. In both cases, NP-related material reaches cell organelles and affects cellular or subcellular functions. The principal mechanism by which NPs evoke cellular response is oxidative stress; thus, the elucidation of cellular uptake and the trafficking of NPs seem to be essential for understanding the mechanisms of ROS generation, especially in the context of interactions of NPs with mitochondria.

The aim of this study was to compare polystyrene NP uptake in three human cell lines of different origin and features. The ability of cells to uptake nanoparticles may vary between cell types of different origin or between cell of similar origin via different functions, e.g., colon goblet cells and enterocytes. Even one type of cell can uptake different nanoparticles by different mechanisms, as was shown in the case of 20 nm silver nanoparticles that were uptaken by microglia via scavenger receptor-mediated mechanisms, whereas CdTe quantum dots were not [12].

A direct comparison of different cell lines in one laboratory on the same batch of material and realized presumably by one group of investigators using the same experimental protocol allows for the elimination of the bias associated with using different laboratories, protocols, and nanoparticles. Thus, in this paper, we studied the mechanisms of the uptake of model MNPlastics, namely polystyrene NPs, by three human cell lines—Caco-2, HT-29, and Hep G2. All three are cancer cell lines, but are derived from different organs. The Caco-2 and HT-29 cells were derived from a human colon and were used here as a primary target for ingested plastic NPs, limiting their entrance to the body. These cells differ in morphology and function, as Caco-2 cells resemble enterocytes, while HT-29 cells mimic goblet cells. Both cell lines are often used in co-culture as a 3D model of the intestines [13]. Hep G2 cells, which exhibit the key characteristics of hepatocytes, were used here as a representative of the main human detoxifying and storing organ—the liver—an organ that plays a significant role in NP accumulation and clearance due to its unique structure and function as a biological filtration system [14]. All three cell lines have been commonly used in a wide range of studies, from oncogenesis to toxicity testing. To minimize the effect of the physical properties of NPs, we used circular NPs of the same density and with similar surface potential (zeta potential) but differing in size. All NPs were prepared using the emulsion polymerization technique to minimize the effect of the protein corona.

Although the mechanisms of NP uptake have been studied for at least two decades and are relatively well understood, any generalizations are still difficult to make due to the variability of nanoparticles and target cellular systems. Therefore, any contribution in the field that allows for the filling of the existing knowledge gaps is a valuable addition.

## 2. Results

### 2.1. Nanoparticle Characterization

In this study, we used fluorescent, uniform NPs of density 1.05 g/cm^3^ from (Magsphere, Inc., Pasadena, CA, USA or NANOCS, Inc., New York, NY, USA) designated by the manufacturer as polystyrene nanoNPs, hereafter designated as polystyrene nanoparticles (PS NPs). According to the respective manufacturer’s technical data, the NPs were prepared using an emulsion polymerization technique; however, the name of the polymerization initiator was not provided. Both manufacturers use 0.1% NaN_3_ as a preservative. The NPs and their characterization by dynamic light scattering (DLS) and Nanoparticle Tracking Analysis (NTA) are provided below in Table 1, Table 2 and Table 3.

The hydrodynamic size of the PS NPs used in this study, as measured by DLS, was much higher than the size claimed by the manufacturers. In contrast, the NTA technique revealed that the size of the NPs was more relevant to that claimed by the manufacturers. This discrepancy is usually attributed to the specificity of the methods, as NTA gives a number-weighted distribution, whereas DLS gives an intensity-weighted distribution and is more sensitive to the presence of aggregates and non-homogeneity of samples. The zeta potential in the culture media of all NPs used in this study was similar, at around negative 30 mV. The zeta potential of the NPs in PBS was slightly higher, at between 30 and 40 mV, with the exception of the 100 nm NPs that were prepared in the presence of an anionic surfactant. Whereas the zeta potential of these NPs in PBS was around −50 mV, in the culture media, it was similar to that of the other NPs, likely due to the buffering capacity of the culture media.

### 2.2. Toxicity of PS NPs

The toxicity of the studied PS NPs was measured by NRU. As shown in Figure 1, the studied NPs were of a low toxicity or non-toxic at all; the reduction of the cell numbers was typically less than 15% of control 48 h after treatment. The most vulnerable were the Hep G2 cells; in this case, the number of living cells was reduced to approximately 75% of control after 24 and 48 h. Interestingly, unlike the other two cell lines, these cells were the most vulnerable to the 100 nm PS NPs. These NPs were prepared in the presence of an anionic surfactant—which might be responsible for the observed effect. Apparently, the Hep G2 cells were the most vulnerable to its action. For the smaller NPs (30 nm and 50 nm) in all cell lines, the number of living cells treated with the lowest concentration of PS NPs (5 μg/mL) was higher than in the control, which suggests the stimulation of cell growth (2–10% after 24 h and 2–7% after 48 h). However, the effect was significant only in the Hep G2 cells. In general, the smallest NPs (30 nm) were more toxic than the larger ones. The two-way ANOVA analysis revealed that the size of NPs was the only factor that affected PS NP toxicity. The investigated cells slightly varied in sensitivity to PS NPs. The Hep G2 cells were the most sensitive, whereas the HT-29 cells were the least sensitive ones.

### 2.3. Uptake of PS NPs of Different Size by HT-29, Caco-2, and Hep G2 Cells

#### 2.3.1. Effect of Size on PS NP Uptake

Using the flow cytometry technique and fluorescent PS NPs, we compared the uptake of NPs of similar size by the studied cell lines. The Caco-2 cells apparently uptook fewer PS NPs than the two other cell lines. The HT-29 and Hep G2 cells uptook the PS NPs with similar effectiveness, despite being of different origin. Regardless of the difference in initial uptake between the Caco-2 cells and the two other cell lines, all three cell lines showed similar regularities in the PS NP uptake. The number of uptaken NPs was highest 6 h after NP addition, except for the Caco-2 cell, where the highest number of NPs was observed after 2 h and then dropped to the lowest values. The smaller NPs were uptaken more readily than the bigger ones (Figure 2). Interestingly, the 50 nm PS NPs were uptaken more effectively than the 100 nm ones, despite the two-fold difference in the actual number of NPs per milliliter (see Table 3).

#### 2.3.2. Uptake of Small and Large PS NPs in Mixture Is Lower than Their Uptake Alone

To further investigate the mechanisms of PS NP uptake, we compared the uptake of the 30 nm and the 100 nm NPs alone and in mixture in one experiment. The uptake of single NPs followed the pattern observed earlier (Figure 2), and this was also true for the uptake of NPs in mixture. However, the uptake of the larger (100 nm) PS NPs in mixture was significantly lower compared with the NPs alone; similarly, the uptake of the smaller ones (30 nm) in mixture was markedly inhibited (Figure 3). A similar pattern was observed in all three lines tested, although the actual number of uptaken PS NPs (fluorescence) differed, as shown in Figure 3.

### 2.4. Effect of Internalization Inhibitors on Uptake of PS NPs by HT-29, Caco-2, and Hep G2 Cells

#### 2.4.1. Effect of Endocytosis Inhibitors on PS NP Uptake

To evaluate the role of a particular internalization mechanism in PS NP uptake, we used different uptake inhibitors. Amiloride, an inhibitor of micropinocytosis, did not affect PS NP uptake in any studied cell line. Nystatin, an inhibitor of caveolin/lipid rafts-dependent internalization, decreased PS NP uptake in HT-29 cells, but to a relatively low extent (87% of the uptake of untreated cells when a higher concentration (100 μg/mL) of PS NPs was used). In contrast, Nystatin markedly increased the uptake of the PS NPs in Caco-2 cells (258% of the uptake of untreated cells when a lower (10 μg/mL) concentration of PS NPs was used). The same effect was observed also for the higher concentration of PS NPs, but to a lower extent (123%). In this cell line, chlorpromazine and Pitstop2, both inhibitors of clathrin-mediated endocytosis (CME) and clathrin-independent, dynamin-dependent endocytosis (FEME), also enhanced PS NP internalization. The effect was appreciable and occurred at both PS NP concentrations (174% and 175% of the uptake of untreated cells for chlorpromazine and Pitstop2 for a 100 μg/mL PS NP concentration, respectively). In contrast, neither chlorpromazine nor Pitstop2 affected PS NP uptake in the HT-29 cells or the Hep G2 cells (Figure 4, Table 4).

#### 2.4.2. Effect of Scavenger Receptor Inhibitors on PS NP Uptake

To further investigate the role of different internalization mechanisms in PS NP uptake, we analyzed the effect of scavenger receptor inhibitors. Since the major differences between the studied cell lines in the mRNA expression of the receptors was between SR-A1 and SR-B1 receptors, we decided to inhibit either of the receptors or both simultaneously (Table 5). Polyinosinic acid, an inhibitor of MSR1 (SR-A1) receptor, inhibited PS NP uptake in all investigated cell lines. The effect was highly pronounced at both tested PS NP concentrations, with the exception of the Caco-2 cells, where the inhibition occurred only in the higher PS NP concentration (92%, 73%, and 78% of the uptake of control cells for the higher PS NP concentration for the HT-29MTX, Caco-2, and Hep G2 cells, respectively). BLT-1, an inhibitor of the SCARB1 (SR-B1) receptor, strongly inhibited the uptake of PS NPs in the HT-29MTX cells at both PS NP concentrations (66% and 81% of the uptake of untreated cells for 10 μg/mL and 100 μg/mL, respectively). The effect was also observed in the Hep G2 cells, but to a lesser extent and only at a the higher NP concentration (89% of the uptake of untreated cells). In the Caco-2 cells, the inhibition of SCARB1 had no effect on PS NP uptake (Figure 5, Table 4). 

Interestingly, inhibition of both receptors resulted in a subtractive effect, i.e., less pronounced than expected. This was especially pronounced when the higher concentration of PS NPs was used. The effect of simultaneous treatment with PIA and BLT was similar to that of BLT alone (Figure 5, Table 4).

To further investigate the mechanisms of PS NP uptake, we inhibited the CMDE/FEME pathway and the SCARB1 receptors. Interestingly, each cell line responded to the simultaneous treatment with CMDE/FEME and SCARB1 receptor inhibition in different ways. In the HT-29 and Hep G2 cells, where the effect of CMDE/FEME inhibition alone was less pronounced, simultaneous inhibition of the CMDE/FEME pathway and the SCARB1 receptors overrode the effect of SCARB1 receptor inhibition alone, as simultaneous inhibition was similar to that of the inhibition of the CMDE/FEME pathway and more pronounced than expected. On the contrary, in the Caco-2 cells, inhibition of CMDE/FEME resulted in a significant increase in PS NP uptake. The effect was the most profound at the PS NP concentration of 10 μg/mL (443% of the uptake of untreated cells) but still visible at the concentration of 100 μg/mL (125% of the uptake of untreated cells). Simultaneous inhibition of the CMDE/FEME pathway and the SCARB1 receptors resulted in a more pronounced than expected inhibition of PS NP uptake (Figure 6, Table 4).

The effects of all inhibitors and their mixtures on the uptake of PS NPs in all three cell lines are graphically summarized in Table 4.

## 3. Discussion

The toxicity of NPs depends on many internal cellular features and physicochemical properties of NPs [22]. At the cellular level, it is nowadays clear that the ability to induce oxidative stress due to the overproduction of free radical species or the failure of antioxidative defense mechanisms are crucial factors in determining NP toxicity [23]. This usually requires direct interaction of NPs with cellular components, e.g., organelles, such as mitochondria, or antioxidant defense enzymes and low molecular weight antioxidants. This direct interaction, in turn, depends on NP internalization and retention; thus, NP uptake is among the main determinants of NP toxicity.

Since cells have different functions in the organism, they also differ in their ability to uptake extracellular material. Solid state materials are usually uptaken by an energy-dependent process known as endocytosis, which is typically divided into two categories: pino- and phagocytosis. Although some evidence suggests that some NPs, especially ones that are very small and are neutral in charge, may enter cells through passive diffusion, it is the energy-dependent processes that are the main mechanisms of NP uptake. Pinocytosis and phagocytosis differ by the size of their endocytic vesicles. During pinocytosis, small particles and/or fluids are transported through small vesicles (a few nanometers to hundreds of nanometers in size), whereas during phagocytosis, large particles are uptaken, and larger vesicles (up to 250 nm) are formed. Pinocytosis can be subcategorized into clathrin-mediated endocytosis (CME), caveolin-mediated endocytosis (CAV), clathrin- and caveolin-independent, dynamin-dependent endocytosis (FEME), and micropinocytosis [24].

In this work, we focused on model nanoplastic particles, namely PS NPs of three different sizes, and evaluated their uptake by three different human cell lines—two epithelial cell lines of gut origin that usually serve as a first contact cells and one cell line of liver origin, a detoxicating/accumulating organ. The nanoparticle uptake-related features of the three cell lines used in this study are summarized in Table 5. Efficiency of the major endocytosis pathways slightly differs between lines. Clathrin-mediated (CMDE)/clathrin-independent dynamin-dependent (FEME) endocytosis is active in all cell lines; however, the HT-29MTX cells seem to be more efficient in CMDE than the other cells [15,16,17,18]. Caveolin/lipid rafts-mediated endocytosis is very efficient in the Caco-2 and HT-29 cells, whereas absent in the Hep G2 cells due to the lack of expression of the CAV1 gene [19]. Macropinocytosis is also effective in all three cell lines [20,21]. The NPs were purchased from two different suppliers in order to make sure that the eventual regularities would be of a general nature.

The toxicity of the studied NPs followed a general rule observed for NPs—the smaller ones were more toxic than the larger ones. This relationship was dose dependent; however, the toxicity of the studied NPs was very low. This is in agreement with already published results, as a majority of them point to low, negligible toxicity of non-modified polystyrene NPs in reasonable concentrations [25,26].

Despite the similar toxicity of PS NPs, the analysis of their uptake revealed significant differences. Using fluorescent PS NPs, we proved significant quantitative differences in their uptake, though a general pattern of the uptake was similar (Figure 2). The smaller NPs were more readily uptaken than the larger ones, regardless of the cell line being tested. This is a general rule that seems to be valid for the majority of NPs [27,28]. Experiments with simultaneous exposure to PS NPs of different size revealed that small PS NPs inhibited the uptake of the larger PS NPs; conversely, the large PS NPs inhibited the uptake of the smaller ones. This observation suggests that, despite the different size, small and large NPs are uptaken by the same or closely related mechanisms. This confirms previously reported results for polystyrene latex bead uptake by baby hamster kidney cells, where the cells uptook fibronectin-coated beads of different size by receptor-mediated endocytosis (phagocytosis). The authors suggested that the critical parameter determining polystyrene bead uptake was the density of the fibronectin molecules on the bead surface rather than the bead size [7].

A possible involvement of scavenger receptors in PS NP uptake was appraised on the basis of the mRNA expression of the receptors. Expression of the *MSR1* gene in Caco-2 cells was 2 times higher than in the HT-29 and Hep G2 cells, whereas expression of the *SCARB1* gene was 2 times higher in the Hep G2 cells than in the HT-29 and Caco-2 cells. Expression of the *CD36* gene was absent in the Caco-2 and HT-29 cells and low in the Hep G2 cells. The mRNA of the *CD68* gene was present in all cell lines, but to a low extent. The *SCARF1* mRNA was present only in the HT-29 cells. None of studied cells expressed the *AGER* gene mRNA (Table 5).

The main difference is that the Hep G2 cells lack expression of the *Cav1* gene and are thus defective in caveolin-mediated endocytosis. The main difference between the Caco-2 cells and the HT-29 cells lies in the two-fold higher expression of the MSR1 gene coding SR-A1 receptor in the Caco-2 cells and in the activity of clathrin-mediated endocytosis (CMDE), which is higher in the HT-29 cells. The results of the experiments using the NP uptake inhibitors are shown in Figure 4 and summarized in Table 4. Surprisingly, in the Hep G2 and HT-29 cells, the PS NP uptake was not sensitive to inhibitors of endocytosis or micropinocytosis, whereas, in the Caco-2 cells, pretreatment with CAV or CMDE/FEME inhibitors resulted in a more intensive uptake of PS NPs.

Since CAV and CMDE/FEME are generally accepted as major routes of NP uptake, their inhibition should impede PS NP uptake, unless there is an alternative way for PS NPs to enter the cell. Indeed, we and others showed that scavenger receptors were involved in the uptake of silver nanoparticles [10,11,12]. To verify whether this mechanism is also involved in the uptake of PS NPs, we treated cells with inhibitors of the MRS1 receptor (polyinosinic acid, PIA) or inhibitors of the SCARB1 receptor (BLT-1) or their combination. Both scavenger receptor inhibitors decreased PS NP uptake in all three lines tested. Furthermore, the combination of both inhibitors also inhibited the uptake of PS NPs, but to a lesser extent than the sum of the effects of the inhibitors alone (expected value). This suggests that there is a competition between receptors and that the same mechanism stands behind MRS1 and SCARB1 receptor-mediated PS NP uptake. Since scavenger receptors class A and class B are associated with lipid rafts [29,30], the deficit of lipid rafts might be a limiting factor of the receptor-mediated uptake of PS NPs.

Furthermore, the SCARB1 receptor is associated with caveolin [31]. Thus, inhibition of caveolin-mediated endocytosis in Hep G2 cells that are deficient in the *Cav1* gene expression and have dysfunctional caveolin-mediated endocytosis should have a negligible effect on PS NP uptake. As expected, inhibition of the SCARB1 (SR-B1) receptor had no effect on PS NP uptake in low concentrations, but surprisingly, it markedly inhibited PS NP uptake in high concentrations, suggesting the existence of an alternative SCARB1 receptor-mediated mechanism of PS NP uptake. Indeed, Wang et al. showed that overexpression of the caveolin-1 protein in the *Cav1* gene-deficient cells did not affect SCARB1-mediated cholesterol ester flux and uptake [32]. The authors suggested that the activity of SCARB1 was independent of caveolin-1. To check whether the activity of the SCARB1 receptor might be associated with clathrin-dependent endocytosis, we decided to inhibit both pathways simultaneously. In the HT-29 and Hep G2 cells, the combined treatment with CMDE/FEME and theSCARB1 receptor had no effect and overrode the effect of inhibition of the SCARB1 receptor alone. The effect of the combined treatment was less pronounced than expected, which suggests competition between these two ways of NP entry. Altogether, this suggests that SCARB1 receptors might also be associated with the CMDE/FEME way of entry, at least in CAV-deficient cells (Hep G2). Similar to the Hep G2 cells, the response of the HT-29 cells to the combined treatment with CMDE/FEME and the SCARB1 receptor inhibitors and the insensitivity of HT-29 cells to the action of CMDE/FEME alone also imply also that CMDE/FEME might be less important in PS NP uptake than suggested by the literature in the case of nanodiamonds [18] or electrotransfection [21]. In the Caco-2 cells, the combination of inhibitors diminished stimulation of PS NP uptake by the CMDE/FEME inhibitor alone, and it seems that the inhibition of CMDE/FEME compensated the inhibition of SCARB1 receptor. The effect was more pronounced than expected and suggested a synergistic action of both inhibitors (Figure 6, Table 4).

Nanoparticle uptake and toxicity present several significant challenges in various fields, including medicine, environmental science, and material science. These challenges reflect the complex interplay of nanoparticle size, shape, surface chemistry, and physicochemical properties, all of which affect NP biodistribution, cellular uptake, and potential toxic effects. A principal mechanism by which NPs activate the cell response is ROS production [33]. Other mechanisms include signaling pathways modulation, cell transduction, and immune modulation [34]. In any case, NP-induced cytotoxicity usually requires the direct interaction of nanomaterial with cellular organelles. Thus, NP cytotoxicity seems to be mutually associated with their uptake. Although it is generally true that higher NP uptake reflects higher toxicity, in this work, we showed that 100 nm NPs were more toxic than 30 nm, despite the much lower uptake of the larger NPs. We explained this phenomenon was likely caused by using a different method of storage/preparation of the 100 nm NPs, as they were supplied in an anionic buffer. Interestingly, the higher toxicity of the 100 nm NPs was observed only in the Hep G2 cells, further underscoring the complexity of NP toxicity but also proving that, despite the intrinsic properties of NPs, their toxicity also depends on cellular context.

Comprehensive attempts for understanding the mechanisms underlying cellular and subcellular interactions in vitro should be performed to provide insights into the effect of NPs in vivo. As the initial step of subcellular NP interaction is the uptake of NPs, a crucial consideration is how NPs interact with cells and their milieu, and how such interactions might cause any toxicity, starting with how NPs interact in transit with cell membranes prior to their interactions with targeted organelles. This information would help us improve the design and synthesis of NPs in order to maximize the clinical benefits while minimizing side effects. Understanding the effects of various NP characteristics on cellular and biological processes will help in the production of NPs that are efficient but also non-toxic. Furthermore, the increasing application of nanomaterials in medicine for diagnosis, treatment, and prevention of various diseases raises safety concerns about their unwanted intrinsic toxicity that may hinder the translation of nanodrugs from bench to clinic. A better understanding of the mechanisms of NP uptake and trafficking should be helpful in recognizing the associated risks of NPs and facilitate successful clinical translation of nanomaterials.

## 4. Materials and Methods

### 4.1. Nanoparticle Characterization

Polystyrene nanoparticles of nominal size 30 nm, 50 nm, and 100 nm were purchased from Magsphere Inc. (Pasadena, CA, USA) and NANOCS, Inc. (New York, NY, USA). The hydrodynamic diameter and zeta potential were measured by dynamic light scattering (DLS) using a Zestasizer Nano ZS system (Malvern, Malvern, UK) at 25 °C with a scattering angle of 173°. PS NP solutions in a full culture medium were diluted 1:8 with PBS and measured in triplicate with 14 sub runs. The suspensions had a pH = 7.4. Zeta potentials were calculated using the Smoluchowski limit for the Henry equation with a setting calculated for practical use (f(ka) = 1.5).

Hydrodynamic diameter and stability of PS NPs in cell culture media were analyzed also by Nanoparticle Tracking Analysis (NTA) using a NanoSight LM10 apparatus with 3.2 Build 16 software (Malvern Instruments, Malvern, UK). PS NPs were incubated in 2 mL cell culture media in tissue culture plates in 10 μg/mL concentration. NTA was performed immediately after the addition of nanoparticles to media (0 h).

### 4.2. Cell Culture

All cells used in this study were purchased from the American Type Culture Collection (ATCC, Rockville, MD, USA) and maintained according to ATCC protocols. Briefly, human hepatic Hep G2 (HB-8065) cells were cultured in EMEM medium supplemented with 10% fetal calf serum (Biological Industries, Haemek, Israel). The HT-29 (HTB-38) cell line is a human epithelial cell line isolated from a white, female patient with colorectal adenocarcinoma. The cell line is often used in 2D and 3D models of the intestinal epithelium. Cells were cultured in DMEM high glucose medium supplemented with 10% fetal calf serum (Biological Industries, Israel). The Caco-2 (HTB-37) cell line is a human epithelial cell line isolated from a white, male patient with colorectal adenocarcinoma. Cells were cultured in EMEM medium supplemented with 20% fetal calf serum (Biological Industries, Israel). All the cells were incubated in a 5% CO_2_ atmosphere at 37 °C.

Uptake-related features of cell lines used in this study together with used inhibitors of endocytosis mechanisms are presented in Table 4.

The inhibitors, their mode of action, and drawbacks of their use are described in detail in several review papers [17,35]. Briefly, chlorpromazine and Pitstop2 were used here as inhibitors of clathrin-mediated endocytosis (CMDE) and clathrin-independent, dynamin-dependent endocytosis (FEME). Both inhibit CMDE, but the mechanism of action is different; therefore, they were used here to confirm the unexpected result of the increased uptake of PS NPs in cells treated with CMDE inhibitors. Chlorpromazine is a drug used to manage and treat schizophrenia, bipolar disorder, and acute psychosis. It also prevents CMDE by anchoring clathrin and the adaptor protein 2 complex to endosomes, thus inhibiting the assembly of coated pits in the plasma membrane [36,37]. Pitstop2 is a rhodamine-based inhibitor of CMDE that binds to the N-terminal domain of clathrin [38]; however, it has also been shown to inhibit clathrin-independent endocytosis [39]. Nystatin, a well-known antifungal antibiotic binds, to cholesterol and inhibits lipid rafts/cholesterol-enriched microdomains/caveolin-mediated endocytosis [40]; however, it is also suspected to interfere with other uptake mechanisms due to the induced changes in membrane fluidity [41]. Amiloride primarily prevents macropinocytosis by inhibition of Na+ channels and Na+/H+ exchange [42] but was also shown to interfere with FEME [43]. Polyinosinic acid is a well-established polyanionic and competitive inhibitor of macrophage receptor 1, whereas BLT-1 (Block Lipid Transporter-1) has been identified as the most potent chemical inhibitor of the SCARB1-mediated selective transfer of lipids [44], though its mechanism of action is still unclear.

### 4.3. Toxicity Testing

Cells were seeded in 96-well microplates (TPP, Trasadingen, Switzerland) at a density of 1 × 10^4^ cells/well in 100 μL of culture medium and left to settle down without treatment for 24 h to obtain optimal cell attachment to the plastic surface. Subsequently, the PS NP suspension was added to the cell cultures to get the final concentration indicated for 24 or 48 h. After incubation, the medium was removed and the cells were washed with 150 μL PBS/well. Then, 100 μL of Neutral Red Assay (working solution 50 μg/mL of the medium) was added to each well and incubated for 3 h. Further, the Neutral Red Assay was removed, the cells were washed with 150 μL PBS/well, and 150 μL of the destaining solution was added to each plate (ethanol/acetic acid/deionized water: 50%/1%/49%). Dye extraction was carried out for 10 min on a microplate shaker, and absorbance of extracted dye was recorded at 540 nm using Infinite200 Pro plate reader spectrophotometer (TECAN, Männedorf, Switzerland).

### 4.4. Flow Cytometry Evaluation of Cellular Uptake of Nanoparticles by Cells

The uptake of PS NPs by the previously mentioned cell lines was examined by flow cytometry. In brief, twenty-four hours after the cells were seeded onto 24-well plates (1 × 10^5^ per well), nanoparticles were added for 2, 6, or 24 h in a 100 μg/mL concentration. After the treatment, cells were washed two times with PBS to remove loosely bound PS NPs, detached with trypsin-EDTA, and spun down. The cell pellet was resuspended in 1 mL PBS for flow cytometry. The analysis was performed on a BD LSR Fortessa cytometer (BD Biosciences, Franklin Lakes, NJ, USA) using an appropriate laser/detector combination. Data from 20,000 events per sample were recorded.

### 4.5. Treatment of Cells by Inhibitors

The inhibitors used in this study were as follows: chlorpromazine, Pitstop2 (N-[5-[(4-bromophenyl)methylene]-4,5-dihydro-4-oxo-2-thiazolyl]-1-naphthalenesulfonamide), Nystatin, Amiloride, polyinosinic acid, BLT-1 (2-(2-hexylcyclopentylidene)-hydrazinecarbothioamide). Cells were treated with the indicated concentration of BLT-1 or Amiloride for 60 min, while with the other inhibitors were used for 30 min. Then, PS NPs were added at indicated concentrations for the next 2 h, and fluorescence was measured by flow cytometry.

The concept of Expected value: Expected value was calculated as a sum of fluorescence of PS alone, net effect of inhibitor 1, and net effect of inhibitor 2:Exp = PS + (Inhib.1 − PS) + (Inhib.2 − PS)(1)

The effect of action of two inhibitors can be additive (both inhibitors act independently; the expected value is similar to the experimental one), subtractive (both inhibitors compete; the expected value is lower than the experimental one), or synergistic (both inhibitors intensify each other’s effect; the expected value is higher than the experimental one).

### 4.6. Statistical Evaluation

All data are presented as mean ± standard deviation, unless indicated otherwise. Three independent experiments were done (*n* = 3) in each case. For toxicity testing, each independent experiment consisted of 4 technical repeats. Significance of difference between means was evaluated by the pairwise *t*-test or the two-way ANOVA. Significance of difference between variances was tested by the F test. Differences with *p* < 0.05 were assumed to be significant. The significance level was statistical evaluation, and graphs were prepared using GraphPad Prism 8 software. To follow the statistical evaluation of the results, a minimal data set is provided as a Appendix A. 

## 5. Conclusions

Thus far, polystyrene NP uptake has generally been ascribed to endocytosis, sometimes distinguishing clathrin-mediated (CMDE) and caveolin-mediated (CAV) endocytosis. In this work, we clearly showed that receptor-mediated endocytosis (RME) is a major pathway of polystyrene NP uptake. Although RME is usually considered a form of CMDE, we clearly point out—inclusively but not exclusively—that two receptors are involved in polystyrene NP uptake but are not coupled with CMDE, namely macrophage receptor 1 (MSR1, SCARA1, CD204) and scavenger receptor class B member 1 (SCARB1, CD36L1). Both receptors are associated with lipid rafts that suggest CAV as a dominant mechanism of polystyrene NP uptake. However, as inhibition of the CAV pathway in the Caco-2 cells resulted in enhanced polystyrene NP uptake, our results also suggest the existence of a yet unidentified compensation mechanism that might be activated when a major mechanism is non-functional. Finally, we showed that even cells of the same origin (colon) might differ substantially in the mechanisms of nanoparticle uptake, which make any generalizations about the mechanisms of polystyrene NP uptake unjustified.

Our results suggest that in vitro attempts for understanding the mechanisms underlying nanoparticle cellular and subcellular interactions are necessary to provide insights into the effect of NPs in vivo. This in turn will help in better understanding the associated risks of NPs and facilitate successful clinical translation of nanomaterials.

## Figures and Tables

**Figure 1 ijms-26-04783-f001:**
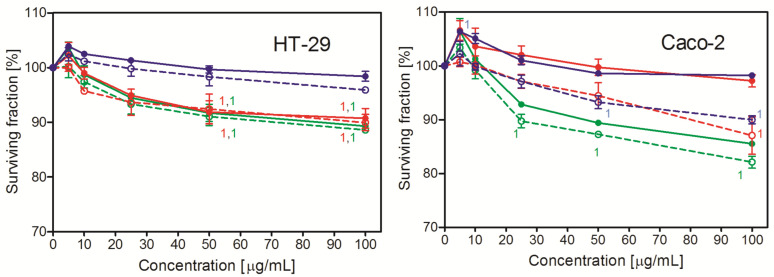
Toxicity of PS NPs in three human cell lines. Nanoparticles were added in indicated concentrations to the cell cultures for 24 h or 48 h, and then the number of cells was estimated by Neutral Red Assay. Mean ± SD, *n* = 3.

**Figure 2 ijms-26-04783-f002:**
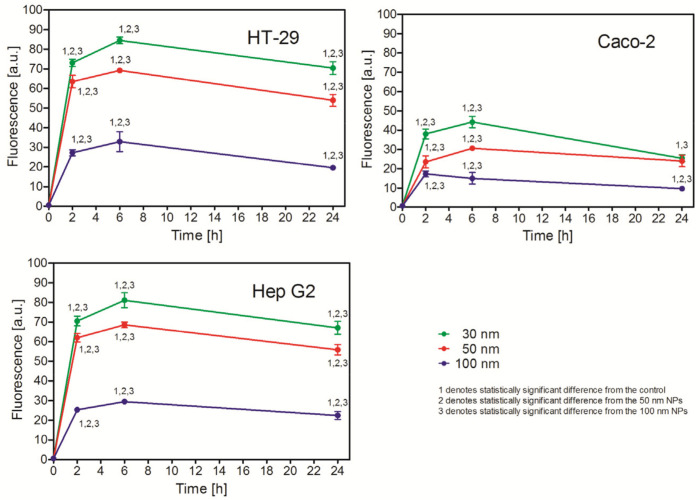
Nanoparticles of different size (100 μg/mL) were added separately for 2, 6, or 24 h, and fluorescence was measured by flow cytometry. Mean ± SD, *n* = 3.

**Figure 3 ijms-26-04783-f003:**
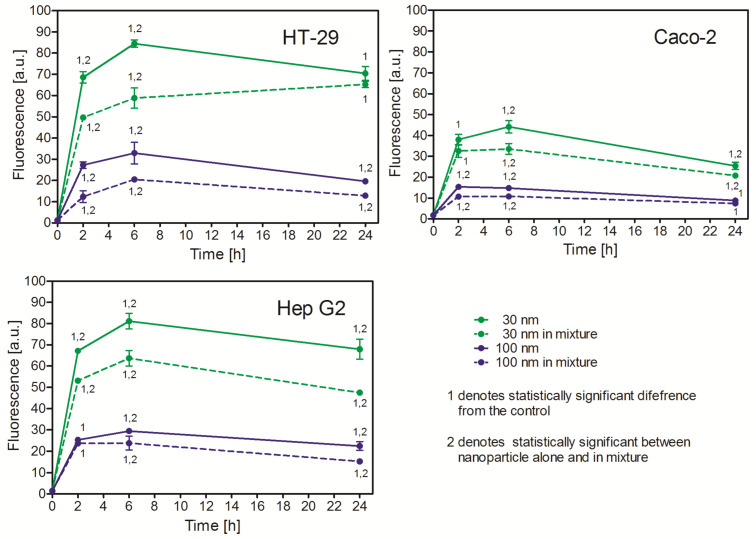
Uptake of 30 nm and 100 nm PS NPs when added alone or in mixture. Both sizes NPs were added in similar concentration (100 μg/mL) for an indicated time. Fluorescence was measured by flow cytometry. Mean ± SD, *n* = 3.

**Figure 4 ijms-26-04783-f004:**
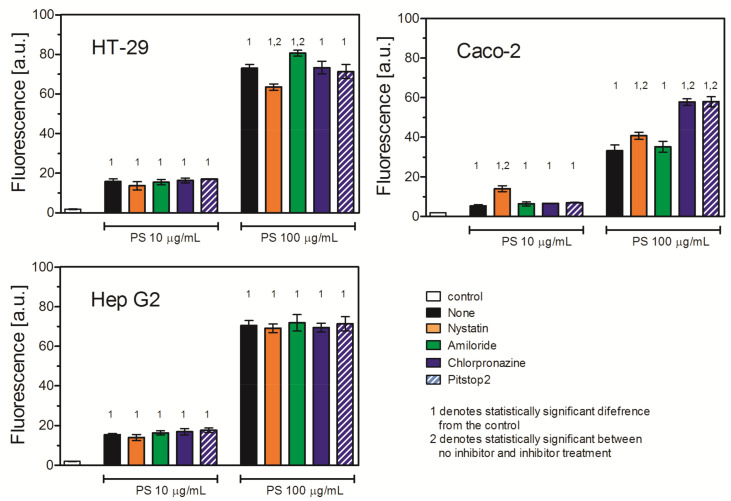
Effect of endocytosis inhibitors on uptake of 30 nm PS NPs. Cells were treated with inhibitors for 30 min, and then 30 nm PS NPs were added for 2 h. Fluorescence was measured by flow cytometry. Mean ± SD, *n* = 3.

**Figure 5 ijms-26-04783-f005:**
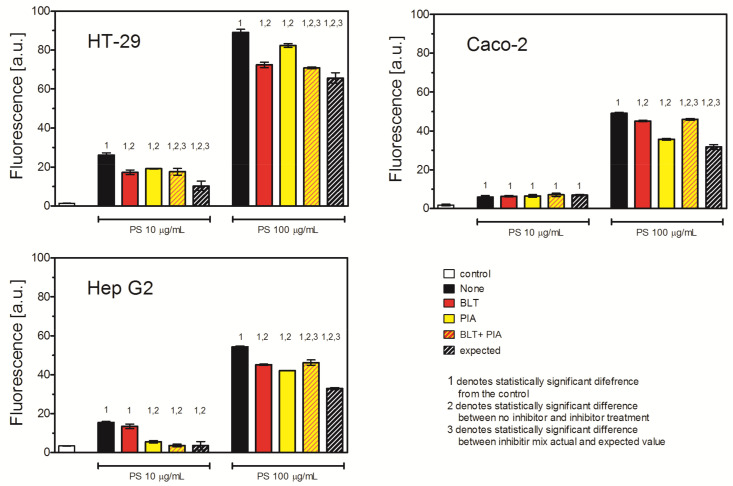
Effect of inhibition of scavenger receptors MSR1 and SCARB1 on uptake of PS NPs. Cells were treated with 20 μM BLT or 10 μg/mL polyinosinic acid (PIA) or their combination for 30 min, and then 30 nm PS NPs were added for 2 h. Fluorescence was measured by flow cytometry. Mean ± SD, *n* = 3. The concept of expected value is explained in Section 4.

**Figure 6 ijms-26-04783-f006:**
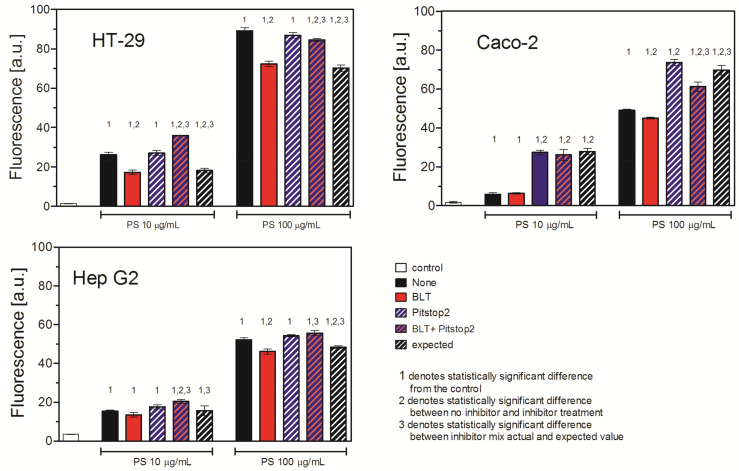
Effect of inhibition of scavenger receptor SCARB1 and clathrin-mediated endocytosis on uptake of PS NPs. Cells were treated with 25 μM Pitstop2 or 20 μM BLT or their mixture for 30 min, and then 30 nm PS NPs were added for 2 h. Fluorescence was measured by flow cytometry. Mean ± SD, *n* = 3. The concept of expected value is explained in Section 4.

**Table 1 ijms-26-04783-t001:** Polystyrene NPs used in this study.

PS NPs	Nominal Size [nm]	Fluorescence	Manufacturer
PS 30	30	Green (FITC), Ex. 498 Em. 517	Magsphere Inc.
PS 50	50	Far-red (Cy5), Ex. 650 Em. 670	NANOCS, Inc.
PS 100	110	Blue, Ex. 331 Em. 395, Anionic surfactant	Magsphere Inc.

**Table 2 ijms-26-04783-t002:** Characterization by DLS of polystyrene NPs used in this study.

PS NPs	Hydrodynamic Size [nm]			
	PBS	HT-29 Medium	Caco-2 Medium	Hep G2 Medium
30 nm	56.96 ± 1.28	34.40 ± 0.44	38.63 ± 1.31	50.59 ± 1.63
50 nm	166.30 ± 2.97	277.16 ± 5.67	36.05 ± 13.85	376.85 ± 15.93
100 nm	95.32 ± 0.39	144.97 ± 0.95	157.23 ± 2.38	170.40 ± 2.07
	Zeta Potential [mV]			
30 nm	−37.66 ± 4.09	−30.56 ± 0.70	−33.13 ± 1.79	−33.30 ± 2.70
50 nm	−36.06 ± 2.90	−32.63 ± 0.20	−33.83 ± 0.66	−36.93 ± 1.95
100 nm	−49.53 ± 5.28	−35.16 ± 0.66	−33.20 ± 0.85	−34.60 ± 1.04

**Table 3 ijms-26-04783-t003:** Characterization by NTA measurements of polystyrene NPs used in this study.

PS NPs	Size [nm]			
	PBS	HT-29 Medium	Caco-2 Medium	Hep G2 Medium
30 nm	35.50 ± 1.61	37.90 ± 2.88	37.33 ± 3.54	32.50 ± 3.90
50 nm	66.20 ± 3.66	67.12 ± 1.44	70.00 ± 3.85	69.99 ± 3.93
100 nm	110.32 ± 2.39	104.88 ± 2.95	107.23 ± 4.55	112.40 ± 4.22
	Concentration [particles/mL]			
	PBS	HT-29 Medium	Caco-2 Medium	Hep G2 Medium
30 nm	5.90 ± 4.64 (×10^14^)	5.80 ± 3.30 (×10^14^)	5.21 ± 5.77 (×10^14^)	5.10∙± 3.22 (×10^14^)
50 nm	4.34 ± 2.89 (×10^12^)	4.90 ± 1.21 (×10^12^)	4.78 ± 3.55 (×10^12^)	5.00 ± 2.10 (×10^12^)
100 nm	6.60∙± 3.45 (×10^14^)	6.33 ± 2.77 (×10^14^)	5.99 ± 1.60 (×10^14^)	6.76 ± 1.54 (×10^14^)

**Table 4 ijms-26-04783-t004:** Effect of endocytosis and scavenger receptor inhibitors on uptake of 30 nm PS nanoparticles.

Process Inhibited	Inhibitor	PS NPs Concentration[μg/mL]	Cell Line
HT-29MTX	Caco-2	Hep G2
Membrane dependent
Clathrin-mediated endocytosis(CMDE/FEME)	Chlorpromazine	10			
100		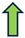	
Pitstop2	10			
100		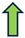	
Clathrin-independent endocytosis (CAV)	Nystatin	10		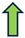	
100	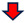	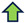	
Micropinocytosis	Amiloride	10			
100	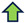		
Scavenger receptor-dependent
MSR1(SCARA1, SR-A1)	Polyinosinic acid	10	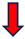		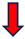
100	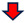	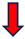	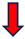
SCARB1(SR-B1)	BLT-1	10	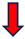		
100	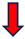	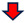	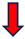
Mixture of inhibitors
Clathrin-mediated endocytosis + SCARB1	Pitstop2+BLT-1	10	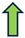	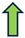	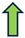
100	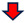	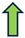	
MSR1 + SCARB1	Polyinosinic acid +BLT-1	10	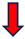		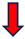
100	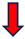	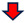	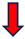

MSR1—macrophage receptor 1 (SCARA1, CD204); SCARB1—scavenger receptor class B member 1 (CD36L1). Upward green arrows mean increase of uptake. Downward red arrows mean decrease of uptake. Size of arrows reflect degree of change (arbitrary). A horizontal rectangle means no effect.

**Table 5 ijms-26-04783-t005:** NP uptake-related features of three cell lines used in this study.

Feature	Caco-2	HT-29MTX	Hep G2	Inhibitor Used in This Study	Lit.
Clathrin-mediated endocytosis (CMDE), clathrin-independent, dynamin-dependent endocytosis (FEME)	+	+++	+	Chlorpromazine, Pitstop2	[15,16,17,18]
Caveolin/lipid rafts-mediated endocytosis, clathrin-independent, (CAV)	+++ CAV1 expression 10× > Hep G2	+++ CAV1 expression 10× > Hep G2	− Very low expression of CAV1	Nystatin	[17,19]
Macropinocytosis	+	++	+	Amiloride	[20,21]
Excerpt from scavenger receptors expression (taken from The Human Protein Atlas, https://www.proteinatlas.org, accessed on 20 November 2024)					
*MSR1* (SCARA1) (consensus name SR-A1)	++ (2× > HT-29	+	+ (similar to HT-29)	Polyinosinic acid (PIA)	
*SCARB1* (consensus name SR-B1)	++ (similar to HT-29)	++	+++ (2× > HT-29	BLT-1	
*CD36* (consensus name SR-B2)	−	−	+	None	
*CD68* (consensus name SR-D1)	+	+	++	None	
*SCARF1* (consensus name SR-F1	−	++	−	None	
*AGER* (RAGE) (consensus name SR-J1)	−	−	−	None	

AGER—advanced glycosylation end product-specific receptor (RAGE); CD36–CD36 molecule (SCARB3); CD68—CD68 molecule (SCARD1); MSR1—macrophage receptor 1 (SCARA1, CD204); SCARB1—scavenger receptor class B member 1 (CD36L1); SCARF1—scavenger receptor class F member 1.

## Data Availability

Data are contained within the article or Appendix A. The original contributions presented in this study are included in the article/Appendix A. Further inquiries can be directed to the corresponding author.

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
