# Peer review of "Uptake and Toxicity of Polystyrene NPs in Three Human Cell Lines"

_ijms, 2025, doi:10.3390/ijms26104783_

Round 1

Reviewer 1 Report

Comments and Suggestions for Authors

The manuscript by Męczyńska-Wielgosz and colleagues sought to investigate the uptake and toxicity of NPs in different human cell lines. This is a short descriptive study.

Here are some minor comments:

  • The authors received their NPs from different manufacturers. Is there any difference in terms of toxicity or intake depending on the manufacturer?
  • The authors used cell lines that have a tendency to generate spheroids. Was the structure of those spheroids affected by the NPs?
  • The rationale for using the selected cell lines is missing in the introduction.
  • 5: The table spreads over 2 pages, rendering it difficult to read. Would it be possible to have it on a single page? What do the rectangular boxes mean?
  • Proofreading needed. For instance, in Fig.5, “difference”, “inhibitir”.
  • Figures 1 and 2: The legends need to be expanded.
  • There are variations in the font size across the manuscript (not all the legends are on the same font size)

Author Response

Answers to the comments of Reviewer 1.

  • The authors received their NPs from different manufacturers. Is there any difference in terms of toxicity or intake depending on the manufacturer?

Yes, indeed, we used polystyrene NPs from two manufacturers. 30 nm and 110 nm NPs were obtained from Megsphere, Inc, CA, USA, whereas 50 nm NPs were obtained from NANOCS, Inc, NY, USA. Despite the negligible toxicity of polystyrene NPs, there were differences in toxicity between them. However we attribute these differences rather to the difference in size of NPs than the differences between manufactures, as the biggest  difference was between 30 nm and 110 nm NPs that were produced by the same manufacturer (Megsphere, Inc, CA, USA). The observed difference in toxicity of NPs followed a general rule known for the other types of NPs: smaller were more toxic than bigger ones.

  • The authors used cell lines that have a tendency to generate spheroids. Was the structure of those spheroids affected by the NPs?

Yes, indeed, all cell lines used in this study can form spheroids. However, these spheroids can be created in special culture condition, such as the hanging drop method or by culturing cells in ultra-low attachment plates. We cultured the cells in the “normal” culture plates, thus the cells grown as a monolayer. Sometimes foci formation was observed, especially in the overgrowth condition, but this problem was mitigated by re-seeding the cells when they reach 70-80 % confluency.  

  • The rationale for using the selected cell lines is missing in the introduction.

The Introduction section was thoroughly rewritten to address the reviewer comments. New Introduction section is presented below, with changes and new additions marked with green.

  1. Introduction

Plastic products that pollute environment slowly disintegrate into small fragments, eventually forming micro- and nanoplastic particles (MNPlastics), i.e. particles less than 5 mm and 1 µm in size, respectively. These tiny plastic particles of various shapes and sizes are widespread in the environment, and can be also found in the human and animal body. Once entered human body, usually by inhalation or digestion [1], the MNPlastics will soon or later find their way to enter cells.

Though internalization mechanisms that are active in a particular cell type likely determine its ability to uptake nanoparticles (NPs), the NPs physicochemical properties may also affect the process. Several NPs features has been proven to affect the internalization process, among them size, shape and surface properties to be of the greatest importance [2, 3]. Furthermore, composition of protein corona, i.e. external proteins that are bound to NPs surface [4], may also affect the NPs uptake.

The common mechanisms of MNPlastics cellular internalization does not differ substantially from the mechanisms of internalization of other types of NPs. Native NPs are uptaken, if they fit a natural processes occurring in the cell, such as uptake of viruses, bacteria or other antigens, or natural compounds, e.g. hormones, growth factors and lipoproteins. Engineered NPs are made to embody natural compounds and merge into those processes, e.g. by linking with ligands with high affinity to highly expressed receptors or proteins secreted in tumor microenvironment, or changing other NPs surface properties, such as charge, hydrophobicity, etc. [5].

In general, energy independent and energy dependent pathways might be considered. The majority of NPs enter cells through energy dependent mechanism, such as phagocytosis and pinocytosis. Whereas phagocytosis is realized mostly by specialized cells, such as neutrophils or macrophages, the latter is available for majority of cells [6]. Depending on the proteins involved in internalization, the pinocytosis is usually further divided into micropinocytosis, clathrin-dependent endocytosis, caveolin dependent endocytosis, clathrin- and caveolin-independent endocytosis. Some mechanisms limit size of uptaken material, e.g. clathrin-coated vesicles are around 100-150 nm in diameter, whereas caveola is usually 60-90 nm in diameter. The uptake of NPs by phagocytosis seems to be size independent, at least for NPs with size up to 1 μm [7]. Thus, NPs properties define the mechanism by which NPs is uptaken.

NPs uptake depends also on target cells. Some specialized cells rely on a particular mechanisms of uptake, e.g. phagocytes, macrophages uptake extracellular material mostly by phagocytosis and micropinocytosis, whereas these mechanism are rarely utilized by other types of cells (for review see [8, 9]). Recently, attention has been paid to the role of receptor mediated endocytosis in the NPs internalization. It is nowadays clear that NPs can interact with scavenger receptors and be uptaken via receptor dependent pathway [10-12]. Thus, it seems that NPs uptake should be considered individually in regards to particular NPs type and cellular system investigated.

NPs uptake and toxicity depend on similar factors suggesting a mutual relationship between the uptake and cytotoxicity of NPs. Once NPs reach plasma membrane they are uptaken, usually by endocytosis, directed via a endosome trafficking network, and sorted to different cellular components. Otherwise, NPs or their content are released into the cytoplasm. In both cases, NPs-related material reaches cell organelles and affects cellular or subcellular functions. The principal mechanism by which NPs evoke cellular response is oxidative stress, thus elucidation of cellular uptake and trafficking of NPs seems to be essential for understanding the mechanisms of ROS generation, especially in the context of interactions of NPs with mitochondria.

The aim of this study was to compare polystyrene NPs uptake in three human cell lines of different origin and features. Cell ability to uptake nanoparticles may vary between cell types of different origin, or between cell of similar origin by different functions, e.g. colon goblet cells and enterocytes. Even one type of cells can uptake different nanoparticles by different mechanisms, as it was shown in case of 20 nm silver nanoparticles that were uptaken by microglia via scavenger receptor mediated mechanisms, whereas CdTe quantum dots were not [12].

Direct comparison of different cell lines in one laboratory, on the same batch of material and realized presumably by one group of investigators using the same experimental protocol allows for elimination of the bias associated with differences between laboratories, protocols and used nanoparticles. Thus, in this paper we studied mechanisms of uptake of a model MNPlastics material, namely polystyrene NPs, by three human cell lines, Caco-2, HT-29 and Hep G2. All three are cancer cell lines, but derive from different organs. Caco-2 and HT-29 cells derive from a human colon and were used here as a primary target for ingested plastic NPs limiting their entrance to the body. The cells differ in morphology and function, as Caco-2 cells resemble enterocytes, while HT-29 mimic goblet cells. Both cell lines are often used in co-culture, as a 3D model of intestine [13]. Hep G2 cells that exhibit the key characteristics of hepatocytes were used here as a representative of the main human detoxifying and storing organ, liver, an organ that plays a significant role in NPs accumulation and clearance due to its unique structure and function as a biological filtration system [14]. All three cell lines were commonly used in a wide range of studies, from the oncogenesis to the toxicity testing. To minimize the effect of physical properties of NPs, we used circular NPs the same density and with similar surface potential (zeta potential), but differing in size. All NPs were prepared by emulsion polymerization technique to minimize the effect of protein corona.

Though, mechanisms of NPs uptake have been studied for at last two decades and are relatively well understood, any generalizations are still difficult to made, due to the variability of nanoparticles and target cellular systems. It makes any contribution in the field a valuable addition that allows for filling of the existing knowledge gaps.

  • 5: The table spreads over 2 pages, rendering it difficult to read. Would it be possible to have it on a single page? What do the rectangular boxes mean?

We shuffled text to make Tab 5 on one page

The rectangular boxes mean „No change”. The explanation is now added below the table.

  • 6: Proofreading needed. For instance, in Fig.5, “difference”, “inhibitir”.

The text was duly spell-checked by third person. All mistakes were corrected.

  • 7: Figures 1 and 2: The legends need to be expanded.

The legends to the Fig. 1 and 2 were expanded.

Was:

Fig. 1 Toxicity of PS NPs in three human cell lines. Mean ± SD, n =3

Is:

Fig. 1 Toxicity of PS NPs in three human cell lines. Nanoparticles were added in indicated concentrations to the cell cultures for 24 h or 48 h and then number of cells was estimated by neutral red assay. Mean ± SD, n =3

Was:

Fig. 2. Uptake of PS NPs of three different sizes by three human cell lines. Mean ± SD, n =3.

Is: Fig. 2. Uptake of PS NPs of three different sizes by three human cell lines. Different size nanoparticles (100 μg/mL) were added separately for 2, 6 or 24 hours and fluorescence was measured by flow cytometry. Mean ± SD, n =3.

  • There are variations in the font size across the manuscript (not all the legends are on the same font size)

Corrected according to the styles format offered by the MDPI provided form.

Reviewer 2 Report

Comments and Suggestions for Authors

The present study by Męczyńska-Wielgosz et al. investigates the uptake mechanisms and toxicity of polystyrene NPs of different sizes in three human cell lines. The study further explores these mechanisms by utilizing various uptake inhibitors to shed light on the processes involved in NP internalization.

The manuscript presents technical details of the study and offers a technical presentation of the results. However, there is limited discussion on the scientific relevance and broader impact of the research conducted. As a result, the manuscript offers limited novelty and originality. To enhance the quality of the manuscript and clarify its broader contributions to the field, several aspects should be further elaborated.

GENERAL COMMENTS:

1. Importance of investigating the uptake mechanisms of NPs, particularly in relation to understanding their toxicity pathways, predicting biological responses, dose and risk assessment should be addressed more comprehensively in the manuscript.

2. The research gaps are currently insufficiently addressed. The Introduction should include a more detailed background information and literature overview to justify the necessity and relevance of the specific research. Providing more contexts on existing studies and highlighting the unique contribution of this work would strengthen the rationale for the study.

3. The aim of the study should be more clearly defined, not just in the context of the specific tasks conducted, but also with respect to the broader objectives.

4. The significance of the study remained insufficiently articulated. It remains unclear if the findings have broader implications, such as for enhancing our understanding of risks in nanotoxicology, contributing to safer nanoparticle development, or addressing issues in occupational and environmental health.

5. Translational potential of the results in insufficiently addressed. It would be helpful to discuss how the findings could be applied to real-world scenarios, such as in the design of safer nanomaterials or in regulatory frameworks for nanoparticle use.

6. The conclusions currently state information that is already well-known and do not reflect the specific findings of this study. The most relevant and specific findings of the present study should be included in conclusions, along with the broader implications of these findings.

SPECIFIC COMMENTS:

1. Sections 4.3., 4.4., 4.5.: Information on the concentrations of NPs used for testing should be added

2. Line 367: Please provide the number of cells seeded per well

3. Section 4.5.: The mechanism of action of the inhibitors used should be explained in more detail to help readers better understand the results and their relevance to the study's aims.

4. Line 379: Please indicate the wavelengths at which the fluorescence was measured

5. Line 380: The phrase “as described above” is unclear in its current context. Please specify to which description this phrase refers.

6. For all assays, information on the number of independent experiments and the number of replicates used in each assay should be provided.

7. Was a solvent control tested in each assay? If so, please provide the results, or explain why a solvent control was not included in the experiments

8. Section 2.4.1 is written in a style more suitable for the Discussion rather than the Results section. It should be rewritten to present only the results, with the discussion of the findings moved to the appropriate section. Furthermore, the information regarding the different uptake properties of the three cell lines could be integrated into the Introduction, once it has been expanded upon.

Author Response

Answers to the comments of Reviewer 2.

Comments and Suggestions for Authors

GENERAL COMMENTS:

  1. Importance of investigating the uptake mechanisms of NPs, particularly in relation to understanding their toxicity pathways, predicting biological responses, dose and risk assessment should be addressed more comprehensively in the manuscript.
  2. The research gaps are currently insufficiently addressed. The Introduction should include a more detailed background information and literature overview to justify the necessity and relevance of the specific research. Providing more contexts on existing studies and
  3. The aim of the study should be more clearly defined, not just in the context of the specific tasks conducted, but also with respect to the broader objectives.
  4. The significance of the study remained insufficiently articulated. It remains unclear if the findings have broader implications, such as for enhancing our understanding of risks in nanotoxicology, contributing to safer nanoparticle development, or addressing issues in occupational and environmental health.
  5. Translational potential of the results in insufficiently addressed. It would be helpful to discuss how the findings could be applied to real-world scenarios, such as in the design of safer nanomaterials or in regulatory frameworks for nanoparticle use.

Thus, understanding the mechanisms of NPs cellular uptake is essential to exploring the biomedical applications of NPs, particularly for drug delivery, while understanding cellular retention of NPs is essential for understanding of efficacy of encapsulated therapeutics

  1. The conclusions currently state information that is already well-known and do not reflect the specific findings of this study. The most relevant and specific findings of the present study should be included in conclusions, along with the broader implications of these findings.

The Introduction and Discussion sections were thoroughly rewritten to address the Reviewer comments. New Introduction and Discussion sections are presented below, with changes and new additions marked with green.

  1. Introduction

Plastic products that pollute environment slowly disintegrate into small fragments, eventually forming micro- and nanoplastic particles (MNPlastics), i.e. particles less than 5 mm and 1 µm in size, respectively. These tiny plastic particles of various shapes and sizes are widespread in the environment, and can be also found in the human and animal body. Once entered human body, usually by inhalation or digestion [1], the MNPlastics will soon or later find their way to enter cells.

Though internalization mechanisms that are active in a particular cell type likely determine its ability to uptake nanoparticles (NPs), the NPs physicochemical properties may also affect the process. Several NPs features has been proven to affect the internalization process, among them size, shape and surface properties to be of the greatest importance [2, 3]. Furthermore, composition of protein corona, i.e. external proteins that are bound to NPs surface [4], may also affect the NPs uptake.

The common mechanisms of MNPlastics cellular internalization does not differ substantially from the mechanisms of internalization of other types of NPs. Native NPs are uptaken, if they fit a natural processes occurring in the cell, such as uptake of viruses, bacteria or other antigens, or natural compounds, e.g. hormones, growth factors and lipoproteins. Engineered NPs are made to embody natural compounds and merge into those processes, e.g. by linking with ligands with high affinity to highly expressed receptors or proteins secreted in tumor microenvironment, or changing other NPs surface properties, such as charge, hydrophobicity, etc. [5].

In general, energy independent and energy dependent pathways might be considered. The majority of NPs enter cells through energy dependent mechanism, such as phagocytosis and pinocytosis. Whereas phagocytosis is realized mostly by specialized cells, such as neutrophils or macrophages, the latter is available for majority of cells [6]. Depending on the proteins involved in internalization, the pinocytosis is usually further divided into micropinocytosis, clathrin-dependent endocytosis, caveolin dependent endocytosis, clathrin- and caveolin-independent endocytosis. Some mechanisms limit size of uptaken material, e.g. clathrin-coated vesicles are around 100-150 nm in diameter, whereas caveola is usually 60-90 nm in diameter. The uptake of NPs by phagocytosis seems to be size independent, at least for NPs with size up to 1 μm [7]. Thus, NPs properties define the mechanism by which NPs is uptaken.

NPs uptake depends also on target cells. Some specialized cells rely on a particular mechanisms of uptake, e.g. phagocytes, macrophages uptake extracellular material mostly by phagocytosis and micropinocytosis, whereas these mechanism are rarely utilized by other types of cells (for review see [8, 9]). Recently, attention has been paid to the role of receptor mediated endocytosis in the NPs internalization. It is nowadays clear that NPs can interact with scavenger receptors and be uptaken via receptor dependent pathway [10-12]. Thus, it seems that NPs uptake should be considered individually in regards to particular NPs type and cellular system investigated.

NPs uptake and toxicity depend on similar factors suggesting a mutual relationship between the uptake and cytotoxicity of NPs. Once NPs reach plasma membrane they are uptaken, usually by endocytosis, directed via a endosome trafficking network, and sorted to different cellular components. Otherwise, NPs or their content are released into the cytoplasm. In both cases, NPs-related material reaches cell organelles and affects cellular or subcellular functions. The principal mechanism by which NPs evoke cellular response is oxidative stress, thus elucidation of cellular uptake and trafficking of NPs seems to be essential for understanding the mechanisms of ROS generation, especially in the context of interactions of NPs with mitochondria.

The aim of this study was to compare polystyrene NPs uptake in three human cell lines of different origin and features. Cell ability to uptake nanoparticles may vary between cell types of different origin, or between cell of similar origin by different functions, e.g. colon goblet cells and enterocytes. Even one type of cells can uptake different nanoparticles by different mechanisms, as it was shown in case of 20 nm silver nanoparticles that were uptaken by microglia via scavenger receptor mediated mechanisms, whereas CdTe quantum dots were not [12].

Direct comparison of different cell lines in one laboratory, on the same batch of material and realized presumably by one group of investigators using the same experimental protocol allows for elimination of the bias associated with differences between laboratories, protocols and used nanoparticles. Thus, in this paper we studied mechanisms of uptake of a model MNPlastics material, namely polystyrene NPs, by three human cell lines, Caco-2, HT-29 and Hep G2. All three are cancer cell lines, but derive from different organs. Caco-2 and HT-29 cells derive from a human colon and were used here as a primary target for ingested plastic NPs limiting their entrance to the body. The cells differ in morphology and function, as Caco-2 cells resemble enterocytes, while HT-29 mimic goblet cells. Both cell lines are often used in co-culture, as a 3D model of intestine [13]. Hep G2 cells that exhibit the key characteristics of hepatocytes were used here as a representative of the main human detoxifying and storing organ, liver, an organ that plays a significant role in NPs accumulation and clearance due to its unique structure and function as a biological filtration system [14]. All three cell lines were commonly used in a wide range of studies, from the oncogenesis to the toxicity testing. To minimize the effect of physical properties of NPs, we used circular NPs the same density and with similar surface potential (zeta potential), but differing in size. All NPs were prepared by emulsion polymerization technique to minimize the effect of protein corona.

Though, mechanisms of NPs uptake have been studied for at last two decades and are relatively well understood, any generalizations are still difficult to made, due to the variability of nanoparticles and target cellular systems. It makes any contribution in the field a valuable addition that allows for filling of the existing knowledge gaps.

  1. Discussion

Toxicity of NPs depends on many internal cellular features and physico-chemical properties of NPs [15]. At the cellular level, it is nowadays clear that ability to induce oxidative stress due to the overproduction of free radical species or failure of antioxidative defense mechanisms is a crucial factor determining NPs toxicity [16]. This usually requires direct interaction of NPs with cellular components, e.g. antioxidant defense enzymes and low molecular weight antioxidants, or organelles, such as mitochondria. This, in turn, depends on NPs internalization and retention, thus NPs uptake is among the main determinants of NPs toxicity.

Since cells have different functions in the organism they also differ in their ability to uptake extracellular material. Solid state materials usually are uptaken by energy dependent process known as endocytosis, that is usually divided into two categories: pino- and phagocytosis. Though some evidence are available that some NPs, especially very small and neutral in charge, may enter cells through passive diffusion, energy dependent processes are the main mechanisms of NPs uptake. Pinocytosis and phagocytosis differ by the size of their endocytic vesicles. During pinocytosis small particles and/or fluids are transported through small vesicles (few nanometres to hundreds of nanometres in size), whereas during phagocytosis large particles are uptaken and larger vesicles (up to 250 nm) are formed. Pinocytosis can be subcategorized into clathrin-mediated endocytosis (CME), caveolae-mediated endocytosis (CAV), clathrin- and caveolae-independent, dynamin dependent endocytosis (FEME) and micropinocytosis [17].

In this work we focused on model nanoplastic particles, namely PS NPs of three different sizes, and evaluated their uptake by three different human cell lines, two epithelial cell lines of gut origin that usually serve as a first contact cells, and cells of liver origin, a detoxicating/accumulating organ. Nanoparticle uptake related features of three cell lines used in this study are summarized in Tab. 5. Efficiency of the major endocytosis pathways slightly differs between lines. Clathrin mediated (CMDE)/clathrin-independent dynamin-dependent (FEME) endocytosis is active in all cell lines, however HT-29MTX cells seems to be more efficient in CMDE that the other cells [18-21]. Caveolin/lipid rafts mediated endocytosis is very efficient in Caco-2 and HT-29 cells, whereas absent in Hep G2 cells due to the lack of expression of CAV1 gene [22]. Macropinocytosis is also effective in all three cell lines [23, 24]. The NPs were purchased from two different suppliers, to make sure that the eventual regularities are of a general nature.

The toxicity of studied NPs followed a general rule observed for NPs, the smaller ones were more toxic than the larger ones. This relationship was dose-dependent, however, toxicity of studied NPs was very low. This is in agreement with already published results, as majority of them point to low, neglectable toxicity of nonmodified polystyrene NPs in reasonable concentrations [25, 26].

Despite the similar toxicity of PS NPs, the analysis of their uptake revealed significant differences. Using fluorescent PS NPs we proved significant quantitative differences in their uptake, though a general pattern of the uptake was similar (Fig. 2). The smaller NPs were more readily uptaken than the larger ones, despite of cell line tested. This is a general rule that seems to be valid for the majority of NPs [27, 28]. Experiments with simultaneous exposure to PS NPs of different size revealed that small PS NPs inhibited the uptake of the larger PS NPs, and opposite the large PS NPs inhibited the uptake of the smaller ones. This observation suggest that despite the different size small and large NPs are uptaken by the same or closely related mechanisms. This confirms previously reported result for polystyrene latex beads uptake by baby hamster kidney cells, where the cells uptaken fibronectin coated beads of different size by receptor mediated endocytosis (phagocytosis). The authors suggested that the critical parameter determining polystyrene beads uptake was density of fibronectin molecules on the bead surface rather than the bead size [7].   

A possible involvement of scavenger receptors in PS NPs uptake was appraised on the basis of receptors’ mRNA expression. Expression of MSR1 gene in Caco-2 cells is 2 times higher than in HT-29 and Hep G2 cells, whereas expression of SCARB1 gene is 2 times higher in Hep G2 cells than in HT-29 and Caco-2 cells. Expression of CD36 gene is absent in Caco-2 and HT-29 cells and low in Hep G2 cells. mRNA of CD68 gene is present in all cell lines, but to low extent. SCARF1 mRNA is present only in HT-29 cells. None of studied cells does not express AGER gene mRNA (Tab. 5).

The main difference is that Hep G2 cells lack of expression of Cav1 gene, thus are defective in caveolin mediated endocytosis. The main difference between Caco-2 cells and HT-29 cells lays in two times higher expression of MSR1 gene coding SR-A1 receptor in Caco-2 cells and activity of clathrin mediated endocytosis (CMDE), that is higher in HT-29 cells. Results of experiment with the use of NPs uptake inhibitors are shown on Fig. 4 and summarized in Tab. 4. Surprisingly, in Hep G2 and HT-29 cells the PS NPs uptake was not sensitive to inhibitors of endocytosis and micropinocytosis, whereas, in Caco-2 cells pretreatment with CAV or CMDE/FEME inhibitors resulted in more intensive uptake PS NPs.

Since, CAV and CMDE/FEME are generally accepted as a major routes of NPs uptake, their inhibition should impede PS NPs uptake, unless there is an alternative way for PS NPs entrance to the cell. Indeed, we and others showed that scavenger receptors were involved in uptake of silver nanoparticles [10-12]. To verify whether this mechanism is also involved in uptake of PS NPs we treated cells with inhibitors of MRS1 receptor (polyinosinic acid, PIA) or inhibitor of SCARB1 receptor (BLT1) or their combination. Both scavenger receptors inhibitors decreased PS NPs uptake in all three lines tested. Furthermore, combination of both inhibitors also inhibited uptake of PS NPs, however to the lesser extent than sum of the effects of inhibitors alone (expected value). This suggests that there is a competition between receptors and the same mechanism stands behind MRS1 and SCARB1receptors mediated PS NPs uptake. Since, scavenger receptors class A and class B are associated with lipid rafts [29,30], the deficit of lipid raft might be a limiting factor of receptor mediated uptake of PS NPs.

Furthermore, SCRAB1 receptor is associated with caveolae [31].Thus inhibition of caveolae mediated endocytosis in Hep G2 cells that are deficient in Cav1 gene expression and have dysfunctional caveolin-mediated endocytosis should have negligible effect on PS NPs uptake. As expected inhibition of SCRAB1 (SR-B1) receptor had no effect on PS NPs uptake in low concentration, but surprisingly it markedly inhibited PS NPs uptake in high concentration, suggesting existence of alternative SCRAB1 receptor mediated mechanism of PS NPs uptake. Indeed, Wang et al. showed that overexpression of caveolin-1 protein in Cav1 deficient cells did not affect SCRAB1 mediated cholesterol ester flux and uptake [32]. The authors suggested that activity of SCRAB1 was independent of caveolin-1. To check whether activity of SCRAB1 receptor might be associated with clathrin dependent endocytosis, we decided to inhibit both pathways simultaneously. In HT-29 and Hep G2 cells the combined treatment with CMDE/FEME and SCRAB1 receptor had no effect and overrode the effect of inhibition of SCRAB1 receptor alone. The effect of combined treatment was lesser than expected one that suggest competition between these two ways of NPs entry. Altogether suggest that SCRAB1 receptors might be also associated with CMDE/FEME way of entry, at least in CAV deficient cells (Hep G2). Similar to Hep G2 cells response of HT-29 cells to the combined treatment with CMDE/FEME and SCRAB1 receptor inhibitors and insensitivity of HT-29 cells to the action of CMDE/FEME alone implies also that CMDE/FEME might be less important in PS NPs uptake then suggested by the literature in case of nanodiamonds [21] or electrotransfection [24]. In Caco-2 cells the combination of inhibitors diminish stimulation of PS NPs uptake by CMDE/FEME inhibitor alone and it seems that inhibition of CMDE/FEME compensate the inhibition of SCRAB1 receptor. The effect was more pronounced than expected that suggested synergistic action of both inhibitors (Fig. 6, Tab. 4).

Nanoparticle uptake and toxicity present several significant challenges in various fields, including medicine, environmental science, and material science. These challenges reflect the complex interplay of nanoparticle size, shape, surface chemistry, and physicochemical properties, all of which affect NPs biodistribution, cellular uptake and potential toxic effects. A principal mechanism by which NPs activate the cell response is ROS production [33]. Other mechanism includes signaling pathways modulation, cell transduction, and immune modulation [34]. In any case, NPs induced cytotoxicity usually requires direct interaction of nanomaterial with cellular organelles. Thus, NP cytotoxicity seems to be mutually associated with their uptake. Although this is generally true that higher NPs uptake reflects higher toxicity, in this work we showed that 100 nm NPs were more toxic then 30 nm despite much lower uptake of the larger NPs. We explained this phenomenon by different method of storage/preparation of 100 nm NPs, as they were supplied in anionic buffer. Interestingly, higher toxicity of 100 nm was observed only in Hep G2 cells, further underscoring complexity of NPs toxicity, but also proving that despite NPs intrinsic properties their toxicity depends also on cellular context.

Comprehensive attempts for understanding the mechanisms underlying cellular and subcellular interactions in vitro should be performed to provide insights into NPs' effect in vivo. As the initial step of subcellular NPs interaction is their uptake, a crucial consideration is how NPs interact with cells and their milieu, and how such interactions might cause any toxicity, starting with how NPs interact in transit with cell membranes prior to their interactions with targeted organelles. This information helps us to improve design and synthesis of NPs to maximize the clinical benefits while minimizing side effects. Understanding the effects of various NP characteristics on cellular and biological processes will help in production of NPs that are efficient but also nontoxic. Furthermore, increasing application of nanomaterials in medicine for diagnosis, treatment and prevention of various diseases emerge the safety concerns about their unwanted intrinsic toxicity that may hinder translation of nanodrugs from a bench to clinic. Better understanding of the mechanisms of NPs uptake and trafficking should be helpful in recognizing NPs associated risk and facilitate successful clinical translation of nanomaterials.

SPECIFIC COMMENTS:

  1. Sections 4.3., 4.4., 4.5.: Information on the concentrations of NPs used for testing should be added

The information of NPs concentration used in the particular experiments were added to the legends of Figures 2 and 3. Figures 4-6 are self-explaining as the NPs concentration is included in the figure.  

  1. Line 367: Please provide the number of cells seeded per well

Done

  1. Section 4.5.: The mechanism of action of the inhibitors used should be explained in more detail to help readers better understand the results and their relevance to the study's aims.

A paragraph describing inhibitors, theirs mechanism of action is now added to Materials and Methods section

  1. Line 379: Please indicate the wavelengths at which the fluorescence was measured

The wavelengths at which the fluorescence was measured were provided in the Tab. 1.

  1. Line 380: The phrase “as described above” is unclear in its current context. Please specify to which description this phrase refers.

The sentence was rephrased as follows:

WAS:

Cells were treated with the indicated concentration of BLT-1 or Amiloride for 60 minutes, while with the other inhibitors for 30 minutes, then PS NPs were added at indicated concentrations for next 2 h. Fluorescence was measured by flow cytometry after indicated time as described above.

IS:

Cells were treated with the indicated concentration of BLT-1 or Amiloride for 60 minutes, while with the other inhibitors for 30 minutes, then PS NPs were added at indicated concentrations for next 2 h and fluorescence was measured by flow cytometry.

  1. For all assays, information on the number of independent experiments and the number of replicates used in each assay should be provided.

All experiment was independently repeated 3 times (n = 3). This is indicated the figure legends. In case of toxicity testing (Neutral red assay) four technical repeats were measured (separate wells on 96-well plate).

Appropriate comment was added to the Material and Methods section.

  1. Was a solvent control tested in each assay? If so, please provide the results, or explain why a solvent control was not included in the experiments

Yes, solvent control (no-cell set-up) was done for each toxicity testing experiment. The measurements of no-cell set-up was subtracted from the measurement of cell containing wells.

  1. Section 2.4.1 is written in a style more suitable for the Discussion rather than the Results section. It should be rewritten to present only the results, with the discussion of the findings moved to the appropriate section. Furthermore, the information regarding the different uptake properties of the three cell lines could be integrated into the Introduction, once it has been expanded upon.

The entire section 2.4.1. was removed from the Results section. Table was moved to Material and methods section to the cell description subsection. The narrative part was moved to Discussion to facilitate the potential reader understanding of results. The Introduction section in our opinion should be more general, giving the reader overall glance on the investigated problem, rather than specific information on uptake mechanisms present in particular cell line. Nevertheless, Introduction section now contains a general description of the cell lines, as required the other Reviewer.

Round 2

Reviewer 2 Report

Comments and Suggestions for Authors

The authors have answered the reviewer’s comments to a considerable extent.

The general comments have been addressed, and efforts have been made to incorporate them. However, the authors did not provide specific responses to each general comment. While the revised Introduction and Discussion have been expanded and now to a certain extent addresses the issues raised, it remains unclear which sections are intended to correspond to which specific comments. Clearer mapping between the reviewer’s feedback and the revisions would be helpful.

One issue from the first-round review remains unresolved. Comment 6 stated: “Conclusions currently state information that is already well-known and do not reflect the specific findings of this study. The most relevant and specific findings of the present study should be included in conclusions, along with the broader implications of these findings.” This point has not been addressed, and the Conclusions section remains unchanged in the revised manuscript. If the authors do not intend to revise this section in accordance with the suggestion, a justification for this decision is needed.

Author Response

Answers to the comments of Reviewer 2.

As required, changes in the manuscript that addressed specific Reviewer’s comment are now mapped to the corresponding comments. Changes to the original version are marked with green.

GENERAL COMMENTS:

  1. Importance of investigating the uptake mechanisms of NPs, particularly in relation to understanding their toxicity pathways, predicting biological responses, dose and risk assessment should be addressed more comprehensively in the manuscript.

In Introduction:

NPs uptake and toxicity depend on similar factors suggesting a mutual relationship between the uptake and cytotoxicity of NPs. Once NPs reach plasma membrane they are uptaken, usually by endocytosis, directed via a endosome trafficking network, and sorted to different cellular components. Otherwise, NPs or their content are released into the cytoplasm. In both cases, NPs-related material reaches cell organelles and affects cellular or subcellular functions. The principal mechanism by which NPs evoke cellular response is oxidative stress, thus elucidation of cellular uptake and trafficking of NPs seems to be essential for understanding the mechanisms of ROS generation, especially in the context of interactions of NPs with mitochondria.

In Discussion:

Comprehensive attempts for understanding the mechanisms underlying cellular and subcellular interactions in vitro should be performed to provide insights into NPs' effect in vivo. As the initial step of subcellular NPs interaction is their uptake, a crucial consideration is how NPs interact with cells and their milieu, and how such interactions might cause any toxicity, starting with how NPs interact in transit with cell membranes prior to their interactions with targeted organelles. This information helps us to improve design and synthesis of NPs to maximize the clinical benefits while minimizing side effects. Understanding the effects of various NP characteristics on cellular and biological processes will help in production of NPs that are efficient but also nontoxic. Furthermore, increasing application of nanomaterials in medicine for diagnosis, treatment and prevention of various diseases emerge the safety concerns about their unwanted intrinsic toxicity that may hinder translation of nanodrugs from a bench to clinic. Better understanding of the mechanisms of NPs uptake and trafficking should be helpful in recognizing NPs associated risk and facilitate successful clinical translation of nanomaterials.

  1. The research gaps are currently insufficiently addressed. The Introduction should include a more detailed background information and literature overview to justify the necessity and relevance of the specific research. Providing more contexts on existing studies and

The common mechanisms of MNPlastics cellular internalization does not differ substantially from the mechanisms of internalization of other types of NPs. Native NPs are uptaken, if they fit a natural processes occurring in the cell, such as uptake of viruses, bacteria or other antigens, or natural compounds, e.g. hormones, growth factors and lipoproteins. Engineered NPs are made to embody natural compounds and merge into those processes, e.g. by linking with ligands with high affinity to highly expressed receptors or proteins secreted in tumor microenvironment, or changing other NPs surface properties, such as charge, hydrophobicity, etc. [5].

In general, energy independent and energy dependent pathways might be considered. The majority of NPs enter cells through energy dependent mechanism, such as phagocytosis and pinocytosis. Whereas phagocytosis is realized mostly by specialized cells, such as neutrophils or macrophages, the latter is available for majority of cells [6]. Depending on the proteins involved in internalization, the pinocytosis is usually further divided into micropinocytosis, clathrin-dependent endocytosis, caveolin dependent endocytosis, clathrin- and caveolin-independent endocytosis. Some mechanisms limit size of uptaken material, e.g. clathrin-coated vesicles are around 100-150 nm in diameter, whereas caveola is usually 60-90 nm in diameter. The uptake of NPs by phagocytosis seems to be size independent, at least for NPs with size up to 1 μm [7]. Thus, NPs properties define the mechanism by which NPs is uptaken.

NPs uptake depends also on target cells. Some specialized cells rely on a particular mechanisms of uptake, e.g. phagocytes, macrophages uptake extracellular material mostly by phagocytosis and micropinocytosis, whereas these mechanism are rarely utilized by other types of cells (for review see [8, 9]). Recently, attention has been paid to the role of receptor mediated endocytosis in the NPs internalization. It is nowadays clear that NPs can interact with scavenger receptors and be uptaken via receptor dependent pathway [10-12]. Thus, it seems that NPs uptake should be considered individually in regards to particular NPs type and cellular system investigated.

Further in the text:

Though, mechanisms of NPs uptake have been studied for at last two decades and are relatively well understood, any generalizations are still difficult to made, due to the variability of nanoparticles and target cellular systems. It makes any contribution in the field a valuable addition that allows for filling of the existing knowledge gaps.

  1. The aim of the study should be more clearly defined, not just in the context of the specific tasks conducted, but also with respect to the broader objectives.

The aim of this study was to compare polystyrene NPs uptake in three human cell lines of different origin and features. Cell ability to uptake nanoparticles may vary between cell types of different origin, or between cell of similar origin by different functions, e.g. colon goblet cells and enterocytes. Even one type of cells can uptake different nanoparticles by different mechanisms, as it was shown in case of 20 nm silver nanoparticles that were uptaken by microglia via scavenger receptor mediated mechanisms, whereas CdTe quantum dots were not [12].

  1. The significance of the study remained insufficiently articulated. It remains unclear if the findings have broader implications, such as for enhancing our understanding of risks in nanotoxicology, contributing to safer nanoparticle development, or addressing issues in occupational and environmental health.

Nanoparticle uptake and toxicity present several significant challenges in various fields, including medicine, environmental science, and material science. These challenges reflect the complex interplay of nanoparticle size, shape, surface chemistry, and physicochemical properties, all of which affect NPs biodistribution, cellular uptake and potential toxic effects. A principal mechanism by which NPs activate the cell response is ROS production [33]. Other mechanism includes signaling pathways modulation, cell transduction, and immune modulation [34]. In any case, NPs induced cytotoxicity usually requires direct interaction of nanomaterial with cellular organelles. Thus, NP cytotoxicity seems to be mutually associated with their uptake. Although this is generally true that higher NPs uptake reflects higher toxicity, in this work we showed that 100 nm NPs were more toxic then 30 nm despite much lower uptake of the larger NPs. We explained this phenomenon by different method of storage/preparation of 100 nm NPs, as they were supplied in anionic buffer. Interestingly, higher toxicity of 100 nm was observed only in Hep G2 cells, further underscoring complexity of NPs toxicity, but also proving that despite NPs intrinsic properties their toxicity depends also on cellular context.

  1. Translational potential of the results in insufficiently addressed. It would be helpful to discuss how the findings could be applied to real-world scenarios, such as in the design of safer nanomaterials or in regulatory frameworks for nanoparticle use.

Comprehensive attempts for understanding the mechanisms underlying cellular and subcellular interactions in vitro should be performed to provide insights into NPs' effect in vivo. As the initial step of subcellular NPs interaction is their uptake, a crucial consideration is how NPs interact with cells and their milieu, and how such interactions might cause any toxicity, starting with how NPs interact in transit with cell membranes prior to their interactions with targeted organelles. This information helps us to improve design and synthesis of NPs to maximize the clinical benefits while minimizing side effects. Understanding the effects of various NP characteristics on cellular and biological processes will help in production of NPs that are efficient but also nontoxic. Furthermore, increasing application of nanomaterials in medicine for diagnosis, treatment and prevention of various diseases emerge the safety concerns about their unwanted intrinsic toxicity that may hinder translation of nanodrugs from a bench to clinic. Better understanding of the mechanisms of NPs uptake and trafficking should be helpful in recognizing NPs associated risk and facilitate successful clinical translation of nanomaterials.

  1. The conclusions currently state information that is already well-known and do not reflect the specific findings of this study. The most relevant and specific findings of the present study should be included in conclusions, along with the broader implications of these findings.

The Conclusion section was revised according to the Reviewer’s suggestions.

WAS:

Our results suggest that receptor mediated endocytosis is the major way of PS NPs uptake in human cells. These results suggest also an existence of alternative compensating mechanisms of uptake that might be activated when a major mechanism is non-functional. Our results suggest that prevalence of a particular uptake mechanisms depends of cellular context, cell type and origin.

IS:

So far, polystyrene NPs uptake was generally ascribed to endocytosis, sometimes distinguishing clathrin-mediated (CMDE) and caveolin-mediated (CAV) endocytosis. In this work we clearly showed that receptor-mediated endocytosis (RME) is a major way of polystyrene NPs uptake. Though, RME is usually considered as a form of CMDE, we clearly point, inclusively but not exclusively, two receptors that are involved in polystyrene NPs uptake, but are not coupled with CMDE, namely macrophage receptor 1 (MSR1, SCARA1, CD204) and scavenger receptor class B member 1 (SCARB1, CD36L1).